# ExpMRC: Explainability Evaluation for Machine Reading Comprehension

**Yiming Cui**[1,2]**, Ting Liu**[1]**, Wanxiang Che**[1]**, Zhigang Chen**[2]**, Shijin Wang**[2,3]

[1]Research Center for SCIR, Harbin Institute of Technology, Harbin, China
[2]State Key Laboratory of Cognitive Intelligence, iFLYTEK Research, China
[3]iFLYTEK AI Research (Hebei), Langfang, China
[1]{ymcui,tliu,car}@ir.hit.edu.cn
[2,3]{ymcui,zgchen,sjwang3}@iflytek.com

## Abstract

Achieving human-level performance on some Machine Reading Comprehension (MRC) datasets is no longer challenging with the help of powerful Pre-trained Language Models (PLMs). However, it is necessary to provide both answer prediction and its explanation to further improve the MRC system's reliability, especially for real-life applications. In this paper, we propose a new benchmark called ExpMRC for evaluating the explainability of the MRC systems. ExpMRC contains four subsets, including SQuAD, CMRC 2018, RACE$^+$, and C$^3$, with additional annotations of the answer's evidence. The MRC systems are required to give not only the correct answer but also its explanation. We use state-of-the-art pre-trained language models to build baseline systems and adopt various unsupervised approaches to extract evidence without a human-annotated training set. The experimental results show that these models are still far from human performance, suggesting that the ExpMRC is challenging.[1]

## 1 Introduction

Machine Reading Comprehension is a task that requires machines to read and comprehend given passages and answer questions and has received wide attention over the past few years. We have seen tremendous efforts to create challenging datasets (Hermann et al., 2015; Hill et al., 2015; Rajpurkar et al., 2016; Lai et al., 2017; Cui et al., 2019; Sun et al., 2020) and design effective models (Kadlec et al., 2016; Cui et al., 2017; Seo et al., 2016).

However, although the state-of-the-art systems can achieve better performance than the average human on some MRC datasets with the help of pre-trained language models (Devlin et al., 2019; Liu et al., 2019; Clark et al., 2020), the explainability of these systems remains uncertain, such as the internal mechanism in neural models and giving text explanations. This raises concerns in utilizing these models in real-world applications. In a realistic view, question answering (QA) or MRC systems that only give final predictions cannot convince the users since these results lack explainability. In this context, Explainable Artificial Intelligence (XAI) (Gunning, 2017) has received much more attention in recent years. XAI aims to produce more explainable machine learning models while preserving high model output accuracy and allowing humans to understand its intrinsic mechanism.

Understanding the intrinsic mechanism of the neural network is a challenging issue. There are several intense discussions on the relevant topics, such as *whether attention can be explanations* (Serrano and Smith, 2019; Jain and Wallace, 2019; Wiegreffe and Pinter, 2019; Bastings and Filippova, 2020). Nonetheless, we could seek post-hoc explainability approaches, which target models that are not

---

[1]Resources are available through `https://github.com/ymcui/expmrc`

readily interpretable by design. Post-hoc approaches resort to diverse means to enhance the model's interpretability (Barredo Arrieta et al., 2020). One of the suitable post-hoc approaches for NLP is to generate text explanations, which is a practical method for alleviating the absence of the neural network's explainability (Cui et al., 2020). Although the text explanation does not necessarily interpret the model's intrinsic mechanism, it is informative to know both the predicted answer and its text explanation, especially for real-life applications.

To better evaluate the MRC model's explainability, in this paper, we propose a comprehensive benchmark ExpMRC for the machine reading comprehension in a multilingual and multitask way, which evaluates the accuracy of both answers and their explanations. The proposed ExpMRC contains four subsets, including SQuAD (Rajpurkar et al., 2016), CMRC 2018 (Cui et al., 2019), RACE$^+$, and C$^3$ (Sun et al., 2020), with additional annotations of the evidence spans, covering span-extraction MRC and multi-choice MRC in both English and Chinese. The MRC model should not only give an answer span or select a choice for the question but also give a passage span as evidence, which creates more challenges. The resulting dataset contains 11K human-annotated evidence spans over 4K questions. The contributions of our paper are as follows.

- We propose a new MRC benchmark called ExpMRC, which aims to evaluate the accuracy of the final answer prediction as well as its explanation.

- We also propose several baseline systems that adopt unsupervised approaches for ExpMRC.

- The experimental results on ExpMRC show that the current pre-trained language models are still far from satisfactory in providing explanations for the predicted answer, suggesting that the proposed ExpMRC is challenging.

## 2  Related Work

Machine reading comprehension has been regarded as an important task to test how well the machine comprehends human languages. In the earlier stage, as most of the models (Dhingra et al., 2017; Kadlec et al., 2016; Cui et al., 2017) are solely trained on the training data of each dataset without much prior knowledge, their performances are not very impressive. However, as the pre-trained language models emerged during these years, such as BERT (Devlin et al., 2019), RoBERTa (Liu et al., 2019), and ELECTRA (Clark et al., 2020), many systems achieved better performances than average humans on several MRC datasets, such as SQuAD 1.1 and 2.0 (Rajpurkar et al., 2016, 2018).

After reaching the 'overhuman' performance, there is another issue to be addressed. The decision process and the explanation of these artifacts remain unclear, raising concerns about their reliability. In this context, XAI becomes more important than ever not only in NLP but also in various directions in AI. However, most cutting-edge systems have been developed on neural networks, and investigating the explainability of these approaches is nontrivial, which is still an ongoing research.

In NLP, some researchers conducted analyses to better understand the internal mechanism of BERT-based architecture. For example, Kovaleva et al. (2019) discovered that there are repetitive attention patterns across different heads in the multi-head attention mechanism indicating its over-parametrization. However, perhaps the most popular discussion is *whether the attention can be explanations*. Some researchers argue that the attention cannot be used as explanations, such as Jain and Wallace (2019) who verified that using completely different attention weights can also achieve the same prediction. In contrast, some works hold positive attitudes about this topic (Wiegreffe and Pinter, 2019; Bastings and Filippova, 2020). These works have brought us different views of attention-based models, but there is still no consensus about this important topic.

In MRC, the most relevant effort in explainability is the creation of HotpotQA (Yang et al., 2018), which is a multi-hop explainable QA dataset. HotpotQA requires the machine to retrieve relevant documents and extract a passage span as the answer along with its evidence sentences. Various models (Qiu et al., 2019; Shao et al., 2020) have been proposed to address this task using supervised learning approaches with labeled training data. However, unfortunately, almost all works focus on achieving higher scores on the benchmark without specifically caring about the explainability. VCR (Zellers et al., 2019) is a multimodal multi-choice question answering dataset, which requires the machine not only to choose a correct answer choice but also to provide a correct rationale via another multi-choice question.

Table 1: Examples in ExpMRC. The evidence of the answer (in passage) is marked with underline. The answer is marked in blue.

| Subset | Passage | Question & Answer |
|---|---|---|
| **SQuAD** | . . . Competition amongst employers tends to drive up wages due to the nature of the job, since there is a relative shortage of workers for the particular position. Professional and labor organizations may limit the supply of workers which results in higher demand and greater incomes for members. Members may also receive higher wages through collective bargaining . . . | **Q**: Who works to get workers higher compensation? 
 **A**: Professional and labor organizations |
| **CMRC 2018** | . . .钩盲蛇（学名："Ramphotyphlops braminus"）是蛇亚目盲蛇科下的一种无毒蛇种，主要分布在非洲及亚洲，不过现在钩盲蛇的分布已推广至世界各地。钩盲蛇是栖息于地洞的蛇种，由于体型细小，加上善于掘洞，因此经常被误认为蚯蚓. . . | **Q**: 钩盲蛇一般生活在什么地形中? 
 **A**: 地洞 |
| **RACE$^+$** | . . .One such plant is the Golden Wattle tree, British scientist David Caneron has found when an animal eats the tree's leaves, the amount of poison increase in the other leaves. "It's like the injured leaves telephoning the others telling them to fight together against the enemy," he said. The tree also sends defense messages to neighboring plants by giving out a special smell. Golden Wattle trees in the nearby 45 meters will get the message and produce more poison within 10 minutes. . . . | **Q**: According to the study, if one Golden Wattle tree is attacked by animals, it can? 
 **A**: tell other trees to protect it 
 **B**: produce more poison within 10 minutes 
 **C**: sent defense messages to the neighboring plants 
 **D**: kill the animals with its leaves |
| **C$^3$** | . . .大学生活是走上社会的预演，可以说，大学里的处世态度和人际关系的成功与否，直接决定着将来在社会上的成败。人是社会性的动物，生活中的每个人都离不开别人的帮助，同时也在帮助着别人。不管是学习、生活、工作，都要求自己要有良好的处理人际关系的能力。一个人要想有良好的人际关系，就要遵循以下几个原则：一是"主动"。要主动和别人交往，主动帮助别人。二是"诚信"。. . . | **Q**: 说话人认为什么因素决定在社会上的成败? 
 **A**: 工作的态度 
 **B**: 朋友的数量 
 **C**: 大学里的学习成绩 
 **D**: 大学里的人际关系 |

Although various efforts have been made, we argue that the explainability is a universal demand for all MRC tasks and different languages but is not restricted to English multi-hop QA. Another issue is that annotating evidence for each task is not feasible, and we should also seek unsupervised approaches, which do not rely on any annotated evidence to minimize the cost.

In this context, we propose ExpMRC to specifically focus on evaluating explainability on four tasks, covering span-extraction and multi-choice MRC in both English and Chinese. ExpMRC does not provide any newly annotated training data. We encourage our community to focus on designing unsupervised approaches to improve the explainability with generalizable approaches for different MRC tasks and even different languages. To the best of our knowledge, this is the first MRC benchmark in a multi-task and multi-lingual setting, which can be used in not only explainability evaluation but also various other directions, such as cross-lingual studies.

## 3 ExpMRC

### 3.1 Subset Selection

The motivation for our dataset is to provide a comprehensive MRC benchmark for evaluating not only the prediction accuracy but also how well it gives for its explanation. Therefore, our dataset is not completely composed of new data. We adopt several well-designed MRC datasets and newly annotated data to form our dataset to minimize the repetitive annotations and place our work in line with previous works. Specifically, our ExpMRC is partly developed from the following datasets, including two span-extraction MRC datasets and one multi-choice MRC dataset.

- **SQuAD** (Rajpurkar et al., 2016) is a well-known dataset for span-extraction MRC. Given a Wikipedia passage, the system should extract a passage span as the answer to the question.

- **CMRC 2018** (Cui et al., 2019) is also a span-extraction MRC dataset but in Chinese. In addition to the traditional train/dev/test split, a challenge set was also released that requires multi-sentence inference while keeping the original span-extraction setting.

- **C$^3$** (Sun et al., 2020) is a Chinese multi-choice MRC dataset. The system should choose a correct option as the answer after reading the passage and question. To ensure domain consistency with other subsets, we only use non-dialogue subsets C$_M^3$.

As the test set of SQuAD is not publicly available, we cannot adopt it directly.[2] Instead, we follow the original dataset construction steps to replicate the subset for testing purposes, where the subset is annotated from English Wikipedia passages. Note that during the subset annotation, we select the passages that do not appear in the original training and development set.

While we can use RACE (Lai et al., 2017) as the $C^3$ counterpart, we decided not to adopt it. We had some in-house collected multi-choice MRC data, which is similar to RACE and is also designed for the middle and high school students in China. More importantly, these data contain additional hints on the answering process, which are very helpful for evidence annotation. Thus, we decided to use our data instead of RACE. We denote this new subset as RACE$^+$.

At this point, we have four subsets (SQuAD, CMRC 2018, RACE$^+$, and $C^3$) to be annotated, containing both span-extraction and multi-choice MRC tasks in both English and Chinese. Note that to preserve the integrity of the test set results, following previous works (Rajpurkar et al., 2016, 2018; Cui et al., 2019), we do not release the test sets to the public.

## 3.2 Annotation Process

All four subsets contain passages, questions, candidates (if applicable), and answers. We only need to annotate their evidence span on top. Before evidence annotation, the annotators are required to consider whether a question is appropriate for annotation. We skipped some questions based on the following criteria.[3]

- Sensitive, offensive, malicious content are not included.

- The evidence span is a simple combination of the question and answer without much syntactical or semantical variance, such as the evidence span being the same or similar to the question text, where the question word is replaced by the answer.

- The questions require external knowledge to be solved and cannot only be inferred from the passage. That is, the evidence should not be formed by passage span.

- The conclusive questions of the whole passage, such as 'what is the best title for this passage?', 'what is the main idea of the passage?', etc. In this situation, the evidence span might be very long.

After the initial check, we begin the evidence annotation process. First, the annotators are asked to read the question and the correct answer (passage span or option text). Because, as the ground truth answer already exists in the original dataset, it is unnecessary to require the annotators to answer the questions again, which increases their burden when they recommend the wrong answer, and they will eventually consult the ground truth answer to find the correct evidence. Then, the annotators select (copy-and-paste) a span from the passage that can be evidence of the answer. The evidence should be a minimal passage span that can support the answer and does not always need to be a complete sentence or clause. We encourage the annotators to select the evidence that needs reasoning skills, although this is not a usual case in these datasets, especially in span-extraction MRC, where most of the questions do not need reasoning.

Selecting a single contiguous span is to make the task much easier to the model, or it will become a sequence labeling task. During the annotation, if a redundant span is included to form a single span, we instructed our annotator that the length of the redundant span should not exceed 30% of the valid span length. However, in most cases (over 90%), a single contiguous span is enough for our selected datasets. It could be problematic for other datasets that require long-range inference, but this does not often happen in our ExpMRC.

The annotators are paid approximately $0.50 per evidence for all types of MRC data. The annotators are either English-majored or Chinese-majored graduate students from China, depending on the dataset language.[4] Additionally, to avoid overworking and decreasing the annotation quality, we set a hard limit on the number of daily annotations for evidence in this project. After reaching a limit of 300 annotations, the system automatically locks and is unlocked the next day.

---

[2]As CMRC 2018 is our previous work, although the test set is not publicly available, we can still use it.

[3]During the initial check, we provide several examples to the annotators for their reference.

[4]The annotators are full-intern students. The cost is only used for estimating total cost of the project.

Following previous works, we also adopt multiple evidence references for each question to maximize the inter-agreement between the annotators. During annotation, we do not reveal the annotated evidence span of the other annotators to the current annotator to increase the diversity and avoid copy-and-paste behavior. After the preliminary annotation, all evidence spans are checked one-by-one to ensure a high-quality dataset. Finally, the annotations are verified that the correct answer can be selected by only reading the evidence and question to ensure that the annotation is valid.

## 3.3 Data Statistics

The statistics of the proposed ExpMRC are listed in Table 2. Note that the 'token' in Table 2 represents the character for Chinese and the word for English. For all subsets, we provide $2 \sim 4$ referential evidence spans for each question. The distribution of the question type in each task's development set is depicted in Figure 1. There are fewer questions of *'who, when, and where'* in RACE$^+$ and C$^3$, suggesting that these subsets are much more difficult.

Table 2: Statistics of the proposed ExpMRC. 'Num.' denotes the number.

|  | SQuAD | | CMRC 2018 | | RACE$^+$ | | C$^3$ | |
|---|---|---|---|---|---|---|---|---|
|  | Dev | Test | Dev | Test | Dev | Test | Dev | Test |
| Language | English | | Chinese | | English | | Chinese | |
| Answer Type | passage span | | passage span | | multi-choice | | multi-choice | |
| Domain | Wikipedia | | Wikipedia | | exams | | exams | |
| Passage Num. | 319 | 313 | 369 | 399 | 167 | 168 | 273 | 244 |
| Question Num. | 501 | 502 | 515 | 500 | 561 | 564 | 505 | 500 |
| Max Answer Num. | 3 | 3 | 3 | 3 | 1 | 1 | 1 | 1 |
| Max Evidence Num. | 2 | 2 | 3 | 3 | 2 | 2 | 4 | 4 |
| Avg/Max Passage Tokens Num. | 146/369 | 157/352 | 467/961 | 468/930 | 311/514 | 324/603 | 426/1096 | 413/1011 |
| Avg/Max Question Tokens Num. | 12/28 | 11/28 | 15/37 | 15/37 | 15/39 | 16/55 | 14/28 | 14/31 |
| Avg/Max Answer Tokens Num. | 3/25 | 3/27 | 6/64 | 5/33 | 6/20 | 6/27 | 7/25 | 7/35 |
| Avg/Max Evidence Tokens Num. | 26/62 | 28/76 | 43/175 | 52/313 | 23/162 | 23/82 | 37/199 | 41/180 |

It should be noted that ExpMRC does not provide any newly annotated training data. We believe there will be a significant improvement in the performance, when there is a proper amount of labeled[5] training data. However, this is not in line with our motivation. We believe that the explainability is within the model but not depend on the labeled training set. We expect our community to develop a self-explainable system and evaluate their generalizability on a multilingual or multitask setting. If these systems generalize well in our dataset, they can be easily applied to other MRC systems with a different task form or language. Also, by developing unsupervised or semi-supervised system will significantly save the cost for annotating evidence text, which is a promising way to develop generalizable and explainable MRC systems.

## 4 Baselines

Given that the proposed ExpMRC is designed to evaluate the explainability in terms of the system's explanation text, we mainly focus on the *unsupervised approaches* for our baseline systems, where ground truth evidence spans are not provided in the training set. We use pre-trained language models as the backbones to generate answers to the questions. Then we apply several methods to generate evidence spans, where we classify them into non-learning and machine learning baselines.

### 4.1 Non-learning Baselines

For non-learning baselines, we mainly use the prediction and question as the clues for finding evidence. For simplicity, we only consider extracting sentence-level evidence in these baselines, although the ground truth evidence may not always be a complete sentence. We first split the passage into several sentences using '.!?' as delimiters. Then we select one of the passage sentences as the evidence prediction. To find more accurate evidence sentences, we adopt three approaches.

---

[5] Specifically, it refers to the ground truth evidence, as the answers are available in each original training set.

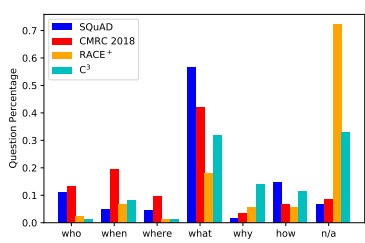

Figure 1: Distribution of question types in each subset.

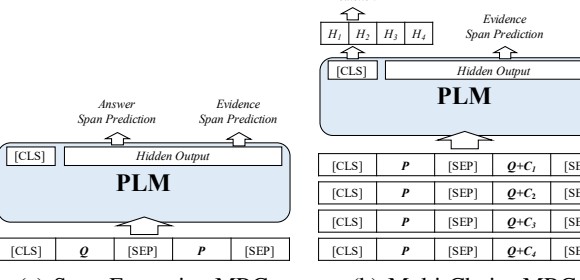

(a) Span-Extraction MRC    (b) Multi-Choice MRC

Figure 2: Architecture of the baseline systems.

- **Most Similar Sentence**: We calculate the token-level F1 score between the predicted answer span (or choice text) and each passage sentence. Then we select the sentence that has the highest F1 as the evidence prediction. In span-extraction MRC tasks, the extracted evidence is the sentence that contains the prediction span in most of the cases.

- **Most Similar Sentence with Question**: Similar to the 'Most Similar Sentence' setting, but we use both the question text and predicted answer span as the key to finding the most similar passage sentence.

- **Answer Sentence**: Particularly, in span-extraction MRC tasks, we can directly extract the sentence that contains the answer prediction as the evidence.

These approaches largely rely on the accuracy of answer prediction, as an incorrect prediction will directly affect the evidence finding process.

## 4.2 Machine Learning Baselines

As no training data are provided in ExpMRC, we seek a pseudo-training approach to accomplish a machine learning baseline system. First, we generate pseudo-evidence for each sample in the respective training set, which has no evidence annotation. We use the ground truth answer and question text to find the most similar passage sentence as the pseudo-evidence to form pseudo-training data. Then we use the pseudo-training data and PLM to train a model that outputs both answer and evidence. Specifically, we add an additional task head on top of the PLM's final hidden representation, alongside its original answer prediction task, as shown in Figure 2.

- **Span-Extraction MRC**: The concatenation of the question $Q$ and passage $P$ are fed into PLM, and we use the final hidden representation with two fully-connected layers to predict the start and end positions of the answer span. The input sequence forms as in Figure 2(b), where `[CLS]` is the special starting token and `[SEP]` is the special token for separation.

- **Multi-Choice MRC**: The concatenation of the passage $P$, question $Q$, and each choice $C_i$ are fed into the PLM to obtain four pooled representations (assuming we have four candidates). Then we use a fully-connected layer with softmax activation to predict the final choice.

The evidence prediction is identical to the answer prediction in span-extraction MRC, where we project the final hidden representation $\boldsymbol{h} \in \mathbb{R}^{n \times h}$ into the start and end probability distributions $p^{\mathrm{s}}, p^{\mathrm{e}} \in \mathbb{R}^n$. Then, we calculate the standard cross-entropy loss of the start and end positions for evidence span prediction.

$$p^{\mathrm{s}} = \mathbf{softmax}(\boldsymbol{h}\mathbf{w}^{\mathrm{s}} + b^{\mathrm{s}}) \, , \; p^{\mathrm{e}} = \mathbf{softmax}(\boldsymbol{h}\mathbf{w}^{\mathrm{e}} + b^{\mathrm{e}}) \tag{1}$$

$$\mathcal{L}_E = -\frac{1}{2N} \sum_{i=1}^{N} (y_i^{\mathrm{s}} \log p^{\mathrm{s}} + y_i^{\mathrm{e}} \log p^{\mathrm{e}}) \tag{2}$$

The final training loss is the sum of answer prediction loss $\mathcal{L}_A$ and the evidence prediction loss $\mathcal{L}_E$, where we apply $\lambda \in [0, 1]$ scaling on $\mathcal{L}_E$, as the pseudo-training data are not quite accurate.

$$\mathcal{L} = \mathcal{L}_A + \lambda \mathcal{L}_E \tag{3}$$

# 5  Evaluation

## 5.1  Evaluation Metrics

To evaluate how well the MRC model can generate explanations for the answers, we use the following metrics, which are divided into answer evaluation and evidence evaluation.

For answer evaluation, we strictly follow the original evaluation script for each subset. Specifically, we use the F1-score (F1) to evaluate SQuAD and CMRC 2018. We discard Exact Match (EM) and only evaluated F1 for simplicity. Note that, as these datasets are in different languages, the evaluation details are slightly different. For RACE$^+$ and C$^3$, we use accuracy for evaluation.

For evidence evaluation, we also use F1 metrics, as most of the evidence spans are quite long, and it is difficult for the machine to extract the evidence spans exactly, and thus we do not adopt EM. Additionally, the central idea of the evidence is to provide enough information to support the answer, so it is proper to adopt F1. Note that we only evaluate the correctness of evidence in this metric, regardless of the correctness of the answer.

Altogether, we also use an overall F1 metric to provide a comprehensive evaluation of the system. For each instance, we calculate the score of the answer metric and evidence metric. The overall F1 of each instance is obtained by multiplying both terms. Finally, the overall F1 of all instances is obtained by averaging all instance-level F1. The overall F1 reflects the correctness of both the answer and its evidence.

$$\mathtt{F1_{overall}} = \mathtt{F1_{answer}} \times \mathtt{F1_{evidence}} \tag{4}$$

## 5.2  Human Performance

Following previous works (Rajpurkar et al., 2016; Lai et al., 2017; Cui et al., 2019), we also report human performance to estimate how well humans perform on this dataset. Following Cui et al. (2019), we use a *cross-validation approach* that regards one of the candidates as prediction and treats the rest of the candidates as ground truths. Final scores are obtained by averaging all possible combinations.

- **SQuAD, CMRC 2018**: In these datasets, there are multiple references for both answer and evidence, and thus we use the cross-validation approach for both and obtain their products as instance-level human performance.
- **RACE$^+$, C$^3$**: As these datasets have only one reference answer, we invite three annotators to answer a random set of 100 questions in each set to obtain the averaged human answer performance. For the evidence, we directly use the cross-validation approach for the selected random set. Similarly, the instance-level human performance is obtained by the product of the answer and evidence score.

Note that as the evidence spans are annotated by referring to either the answers or additional hints, the actual human performance can be lower, and thus, these results should be regarded as *ceiling* human performance roughly. Finally, we average the scores in all instances to obtain the final overall human performance. Note that the answers and the evidences are not annotated by the same annotator, where the former is from the original dataset and the latter is ours.

# 6  Experiments

## 6.1  Setups

We use pre-trained language models as the baseline system backbones. Specifically, we use BERT-base and BERT-large-wwm (Devlin et al., 2019) for English tasks, and MacBERT-base/large (Cui et al., 2020) for Chinese tasks. We use a universal initial learning rate of 3e-5 and iterate two training epochs for all tasks. The maximum sequence length is set to 512, and the QA length is 128 in all

Table 3: Baseline results on SQuAD, CMRC 2018, RACE$^+$, and C$^3$. 'Sent.' for 'sentence', 'Ques.' for 'question'. 'Ans.', 'Evi.', and 'All' denote answer/evidence/overall score, respectively.

| System | SQuAD (dev) | | | SQuAD (test) | | | CMRC 2018 (dev) | | | CMRC 2018 (test) | | |
|---|---|---|---|---|---|---|---|---|---|---|---|---|
| | Ans. | Evi. | All | Ans. | Evi. | All | Ans. | Evi. | All | Ans. | Evi. | All |
| *Human Performance* | *90.8* | *92.1* | *83.6* | *91.3* | *92.9* | *84.7* | *97.7* | *94.6* | *92.4* | *97.9* | *94.6* | *92.6* |
| *PLM Base-level Baselines* | | | | | | | | | | | | |
| Most Similar Sent. | **87.4** | 81.8 | 74.5 | 87.1 | 85.4 | 76.1 | **82.3** | 71.9 | 60.1 | 84.4 | 62.2 | 52.9 |
| Most Similar Sent. w/ Ques. | **87.4** | 81.0 | 72.9 | 87.1 | 84.8 | 75.6 | **82.3** | 76.9 | 63.9 | 84.4 | **69.8** | **59.9** |
| Predicted Answer Sent. | **87.4** | 84.1 | **76.4** | 87.1 | 89.1 | **79.6** | **82.3** | 78.0 | 66.8 | 84.4 | 69.1 | 59.8 |
| Pseudo-data Training | 87.0 | 79.5 | 70.6 | **88.0** | 78.6 | 69.8 | 81.5 | 73.2 | 60.4 | **85.9** | 61.3 | 52.4 |
| *PLM Large-level Baselines* | | | | | | | | | | | | |
| Most Similar Sent. | **93.0** | 83.9 | 79.3 | 92.3 | 85.7 | 80.4 | 82.8 | 71.6 | 60.3 | 88.6 | 63.0 | 55.9 |
| Most Similar Sent. w/ Ques. | **93.0** | 81.9 | 77.4 | 92.3 | 85.1 | 79.8 | 82.8 | 76.3 | 63.6 | 88.6 | **71.0** | 63.2 |
| Predicted Answer Sent. | **93.0** | **85.4** | **81.8** | 92.3 | 89.6 | 83.6 | 82.8 | **77.7** | 66.9 | 88.6 | 70.6 | **63.3** |
| Pseudo-data Training | 92.9 | 80.7 | 75.6 | **93.9** | 80.1 | 74.8 | **83.8** | 73.1 | 62.7 | **89.6** | 62.9 | 55.3 |

| System | RACE$^+$ (dev) | | | RACE$^+$ (test) | | | C$^3$ (dev) | | | C$^3$ (test) | | |
|---|---|---|---|---|---|---|---|---|---|---|---|---|
| | Ans. | Evi. | All | Ans. | Evi. | All | Ans. | Evi. | All | Ans. | Evi. | All |
| *Human Performance* | *92.0* | *92.4* | *85.4* | *93.6* | *90.5* | *84.4* | *95.3* | *95.7* | *91.1* | *94.3* | *97.7* | *90.0* |
| *PLM Base-level Baselines* | | | | | | | | | | | | |
| Most Similar Sent. | 62.4 | 36.6 | 28.2 | 59.8 | 34.4 | 26.3 | 68.7 | 57.7 | **47.7** | 66.8 | 52.2 | 41.2 |
| Most Similar Sent. w/ Ques. | 62.4 | 44.5 | 31.5 | 59.8 | 41.8 | **27.3** | 68.7 | **62.3** | 47.3 | 66.8 | 57.4 | **42.3** |
| Pseudo-data Training | **63.6** | **45.7** | **31.7** | **60.1** | **43.5** | 27.1 | **70.9** | 59.9 | 43.5 | **69.0** | **57.5** | 40.6 |
| *PLM Large-level Baselines* | | | | | | | | | | | | |
| Most Similar Sent. | **69.0** | 37.6 | 29.9 | 68.1 | 36.8 | 28.9 | 73.1 | 59.4 | 49.9 | 72.0 | 52.7 | 43.9 |
| Most Similar Sent. w/ Ques. | **69.0** | **48.0** | **36.8** | 68.1 | **42.5** | **31.3** | 73.1 | 63.2 | **50.9** | 72.0 | 58.4 | 46.0 |
| Pseudo-data Training | **69.0** | 45.9 | 32.6 | **70.4** | 41.3 | 30.8 | **76.4** | **64.3** | 50.7 | **74.4** | **59.9** | **47.3** |

experiments. We use ADAM Kingma and Ba (2014) with weight decay optimizer for training. All experiments are performed on a single Cloud TPU v2 for base-level PLMs and v3 for large-level PLMs. We set $\lambda = 0.01$ for span-extraction tasks and $\lambda = 0.1$ for multi-choice tasks in the final loss function to penalize the evidence pseudo-data training, which we found to be effective. Further investigation is discussed in Section 6.3.

## 6.2 Baseline Results

The results are in Table 3, where 5-run maximum scores are reported. Overall, the best-performing baselines are still far behind the human performance, indicating that the proposed dataset is challenging. Additionally, the gaps in multi-choice MRC subsets are larger than those in span-extraction MRC. For all subsets, adding question text for similarity calculation is more effective than only using the predicted answer. For span-extraction MRC, traditional token similarity methods seem to be more effective as the answer is already a passage span, and its evidence often lies around its context. In contrast, the pseudo-data training approach is more effective in multi-choice MRC, where the options are not composed of the passage span, which is not capable of direct mapping, and it requires similarity calculation in semantics but not only in the token-level calculation.

Improving both answer and evidence prediction does NOT necessarily improve the overall score. For example, in the C$^3$ development set, pseudo-data training at a large-level baseline yields better performance on both answer and evidence prediction than the others. However, its overall score of 50.7 is lower than the best-performing baseline of 50.9. After checking the prediction file, we discovered that there are more samples that have either better evidence spans for the wrong answer prediction or worse evidence spans for correct answer prediction, which decreases the overall score.

Another interesting observation is that although pseudo-data training baselines do not yield better overall scores mostly, we see almost consistent improvements in the answer prediction accuracy, such as in C$^3$ using large-level PLM (e.g., dev +3.3, test +2.4). This suggests that using pseudo evidence helps improve answer prediction, and we expect there will be another improvement when we use a more effective method for extracting high-quality pseudo evidence.

## 6.3 Answer and Evidence Balance

To balance the ratio between the answer and evidence loss, we apply a lambda term on the evidence loss. To explore the effect of the lambda term, we select different $\lambda \in [0, 1]$ and plot the 5-run average dev performance of each task using base-level PLMs. The results are shown in Figure 3.

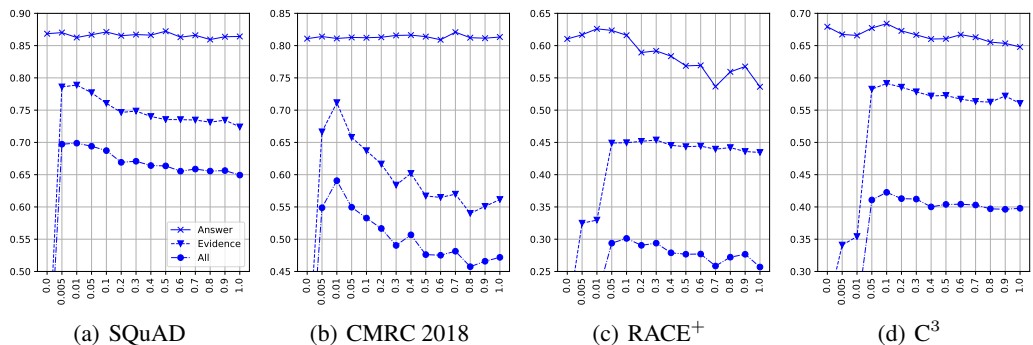

|  (a) SQuAD | (b) CMRC 2018 | (c) RACE$^+$ | (d) C$^3$ |

Figure 3: Effect of the lambda term in the evidence loss. X-axis: lambda, Y-axis: average F1.

Overall, as can be seen, by increasing the lambda term, the evidence score and overall score decrease, suggesting that the pseudo-data training cannot be regarded as important as the original supervised task training (answer prediction), as the pseudo-data are not constructed by the ground truth evidence. However, in regard to the answer score, we observe that the span-extraction MRC tasks are less sensitive to the lambda term than the multi-choice MRC tasks.

The optimal lambda value differs in span-extraction and multi-choice MRC tasks, where SQuAD and CMRC 2018 show smaller value than RACE$^+$ and C$^3$. A possible guess is that two subtasks (answer extraction and evidence extraction) are the same in span-extraction MRC, and thus, the evidence extraction task benefits from the learning of answer extraction. However, as the evidence labels are not accurate enough, increasing lambda term hurts the learning of evidence extraction.

## 6.4 Upper Bound Test for Evidence Extraction

In this section, we analyze the possible steps to achieve better evidence extraction performance. In addition to the 'Most Similar Sentence with Question' and 'Predicted Answer Sentence' (PA Sent.), we also provide two additional baselines for large-level PLMs. We extract the sentence that contains the ground truth answer (GA Sent.) and evidence (GE Sent.) to measure the upper bounds for those systems that only extract sentence-level evidence. The results are shown in Table 4.

Table 4: Upper bound performance of evidence F1 on the development sets.

|  | SQuAD | CMRC 2018 | RACE$^+$ | C$^3$ |
|---|---|---|---|---|
| Most Similar Sent. w/ Ques. | 81.9 | 76.3 | 48.0 | 63.2 |
| Predicted Answer Sent. | 85.4 | 77.7 | - | - |
| Ground Truth Answer Sent. | 88.2 | 82.1 | 49.9 | 66.8 |
| Ground Truth Evidence Sent. | 91.6 | 85.2 | 86.9 | 89.1 |
| *Human Performance* | *92.1* | *94.6* | *92.4* | *95.7* |

As can be seen, the PA-GA and GA-GE gaps in span-extraction MRC are very small (approximately 3%~5%), suggesting that the current system is about to reach the ceiling performance when only using sentence-level evidence extraction. In contrast, in multi-choice MRC, we see a large gap between GA and GE, indicating that only using the answer sentence is not enough to achieve strong evidence extraction performance.

The gap between GE and human performance indicates the gains from expanding sentence-level evidence to a free-form evidence span. In addition to the SQuAD task, the others yield a 5.5%~9.4%

gap, which demonstrates that finding the exact evidence span in these tasks can still achieve a decent improvement.

## 7   Conclusion

In this paper, we propose a comprehensive benchmark for evaluating the explainability of machine reading comprehension systems. The proposed ExpMRC benchmark contains four datasets, covering span-extraction MRC and multiple-choice MRC in both English and Chinese. ExpMRC aims to evaluate the MRC system to give not only correct predictions on the final answer but also extract correct evidence for the answer. We set up several baseline systems to thoroughly evaluate the difficulties of ExpMRC. The experimental results show that both traditional and state-of-the-art pre-trained language models still underperform human performance by a large margin on most of the subsets, indicating that more efforts should be made on designing effective approach for evidence extraction. We hope the release of the dataset will further accelerate the research of explainability and interpretability of MRC systems, especially for the unsupervised approaches.

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
