# OpenReview forum: "ExpMRC: Explainability Evaluation for Machine Reading Comprehension"
_NeurIPS.cc/2021/Track/Datasets_and_Benchmarks/Round1 — Submitted to NeurIPS 2021 Datasets and Benchmarks Track (Round 1)_

### Official Review · Reviewer_f5PG · 2021-07-04
**Useful data annotations to study how interpretable MRC systems are, but the paper lacks clarity**

**Rating:** 5
**Confidence:** 2

**Strengths:**

-	The data set annotations provide a step towards developing unsupervised learning systems that perform self-explanatory predictions; as such I think that it is relevant to the Chinese and English-speaking NLP communities.
-	Baselines are provided.


**Weaknesses:**

- The paper lacks clarity, which makes reading the paper cumbersome: it is sometimes the reader’s task to guess what the authors intended to say. I will provide examples in the Sections below.

**Additional Feedback:**

I am currently tending to a 'weak reject' of the paper. If the authors could improve the clarity of the current manuscript, I would consider chaning my score.

**Clarity:**

When you say 'gold evidence', do you mean 'ground-truth'? (Section 4, Section 6.3 and footnote 5)

In the caption of Figure 2 you do not explain what the following terms stand for:
 - CLS, SEP, P, Q (I could also not find them in the main text)

There is little explanation provided in the caption of Table 2, which is explained using 1 sentence in the main text.
 - 'max answer #' and 'max evidence #' are not explained

There is little explanation provided in the caption of Figure 3

In Section 4.1 you introduce your 'Non-Learning Baselines'. What does non-learning mean in this context? The following things are confusing me:
- You say 'We calculate the token-level F1 score between the predicted answer span (or choice text) and each passage sentence.' When you say 'predicted answer span', where does the prediction come from? From the unsupervised learning algorithm? But that would entail that some learning is happening?! Or is the prediction simply a randomly selected passage that you choose as your prediction? This has not become clear to me from Section 4.1.




**Correctness:**

The data set annotations seem to be constructed in a sensible way:
 - to ensure high-quality annotations, the authors made sure that the human annotators did not complete more than 300 annotations per day. How was this number determined? Why was it not set to 100 or 150 annotations?
 - for the English and Chinese language data sets the annotators were English or Chinese majored graduate students.

However, some asepcts within Section 3.2 remain unclear to me:
- How was the number of maximum annotations determined? Why was it not set to a smaller number such as 100 or 150 annotations?
- How do you combine the annotations of different students into one annotation? What do you do, if students disagree on the annotations? How often does this happen?

**Documentation:**

There exists sufficient detail on the availability and responsible use of the data sets and baselines.

**Ethics:**

There are no ethical concerns.

**Relation To Prior Work:**

The relation to previous work on Machine Reading Comprehension (MRC) systems is discussed. It is not discussed whether there exists any related work that also provides human-annotated texts to study the explainbility of MRC systems - it would be great if the authors could provide some context on this.

**Summary And Contributions:**

The authors provide data set annotations that help understand whether large language models achieve human-level performance on Machine Reading Comprehension tasks for the right reasons (i.e., correct explanations). In addition, the authors provide unsupervised baseline models that extract explanations without using human annotations.

---

### Official Review · Reviewer_1DKa · 2021-07-05
**A new benchmark for Explainable Reading Comprehension**

**Rating:** 6
**Confidence:** 4
**Correctness:** The baselines and evaluation protocol…
**Clarity:** The paper is clear and easy to follow.

**Strengths:**

1. This is a well-written paper and it fills the explainability issues of neural MRC models.
2. The experiment settings and baselines are appropriate and well executed.

**Weaknesses:**

1. In terms of the practical values of explanation, the predicted evidence can serve as a proxy for the uses to validate the predicted answer. It is unknown how much additional value this kind annotation could bring compared to simply producing the answer sentence.
2. While the authors claim that an unsupervised setting is beneficial for generalization, I am not convinced that model trained on this kind of supervision will not be useful. Also, since the model does not have access to the format of the desired evidence (i.e., length, span or sentence), an unsupervised setting might underestimate existing models.

**Additional Feedback:**

N/A

**Documentation:**

Public link is available in the paper.

**Relation To Prior Work:**

Compared to existing work that also study evidence prediction, this work does not assume training annotations of the evidence.

**Summary And Contributions:**

This work adds additional minimal evidence annotation to several existing datasets. In contrast existing settings where the supporting evidence is also provided in the training set, this work presents an unsupervised setting. Baselines using pretrained models are still away from human performance on 3 of the 4 datasets.

---

### Official Review · Reviewer_S6U4 · 2021-07-05
**Good motivation but rough design and implementation**

**Rating:** 6
**Confidence:** 3
**Correctness:** The construction of the dataset is so…
**Clarity:** The paper is well written and easy to…

**Strengths:**

The dataset is well-motivated. The explainability of deep models (especially the ones that have achieved excellent performances in various NLP/CV tasks) is an interesting and important topic.

The experiments are clear and informative (even though I have some complaints about the design of the dataset).

The ExpMRC dataset covers English and Chinese MRC, as well as span-extracting and multi-choice MRC, which should be appreciated.

**Weaknesses:**

The major problem is that, based on the paper, it is not convincing that the proposed dataset really helps assess/improve the models' explainability. Given the results presented in Table 3&4, it seems that the "evidence" task is trivial in span-extraction MRC, as in almost 90% of the cases the evidence is simply the sentence containing the answer. Based on my own experience working on RACE, the "evidence" is either trivial (can be retrieved using heuristics), or not meaningful (answering the question requires complex logical reasoning) for multi-choice MRC tasks. I was eager to read analysis/discussion/examples of this issue, but the examples presented in the paper are all trivial cases. It is important to show how the answer can be inferred from non-trivial evidence sentences, so that we can understand how the ExpMRC dataset measures a models' explainability.

The pseudo-data training baseline is not quite meaningful, as the model is basically trained to learn the heuristic rules. Its performance on producing answers is predictable, because the model has access to ground-truth hints, while its performance on producing evidence is not informative given the results of the non-learning baselines.

A minor weakness, this dataset is very similar to the retrieval task of HotpotQA, limiting the novelty of this dataset.


**Additional Feedback:**

While I have several negative comments about this paper, I still think this is an interesting and important topic, and I would love to hear the authors' feedback about the weaknesses section.

**Documentation:**

The proposed dataset is well documented.

**Relation To Prior Work:**

Most important related work is covered. The high-level concept of explaining models' choice is similar to the VCR task [1] (which is an image-language task), and it should be discussed in the paper.

[1] Zellers, Rowan, et al. "From recognition to cognition: Visual commonsense reasoning." Proceedings of the IEEE/CVF Conference on Computer Vision and Pattern Recognition. 2019.

**Summary And Contributions:**

This paper proposes an explainable machine reading comprehension dataset, ExpMRC, by requiring the system to produce an evidence span from the input document that supports the answer predicted by the system. The proposed ExpMRC dataset is based on previous MRC datasets SQuAD, CMRC, C^3, and a modified version of RACE, and further human annotation is conducted on a subset of their test sets to mark the evidence span given the ground-truth answer. Heuristic baselines that make direct use of the question text/predicted answer/gold answer, as well as baseline models trained on these heuristically retrieved text are tested on the ExpMRC dataset.


=====================post rebuttal=====================

The authors' rebuttal partly addresses my concerns. If the authors provide a more detailed analysis of their dataset as promised in their responses, I think this dataset will be more convincing and worth publishing.

Rating changed 5=>6

---

### Author Response · Authors · 2021-07-09
**Response to All Reviewers**

We are grateful that PCs allow authors to use one additional page to improve their papers based on the reviewers' suggestions.

Again, we would like to thank all reviewers for their constructive comments on our paper. We have revised our paper according to the reviewers' suggestions. Primarily,

1. [*Reviewer S6U4*] We have added an explicit discussion on VCR in Section 2. Please check line 82 in our paper.
2. [*Reviewer f5PG*] We have improved the clarity issues according to your suggestions. The modifications appear in line 212-219, the caption and content of Table 2, the caption of Table 3. Also, several terms are updated, such as 'gold'. Regarding the term 'non-learning', it was illustrated in line 184-186. Please let us know if it is still not clear, and we are happy to revise it.
3. We add line numbers in our paper for quick reference.

If there is anything that should be revised further, please let us know at any time. Thank you.

---

### Decision · Program_Chairs · 2021-07-26

**Decision:**

Reject

**Comment:**

Overall, the paper presents an interesting idea but is with a relatively rough design. All reviewers make marginal scores. The current version will require a series of additional work to make it ready for publication.